# DISENTANGLED REPRESENTATION LEARNING WITH INFORMATION MAXIMIZING AUTOENCODER

## ABSTRACT

Learning disentangled representation from any unlabelled data is a non-trivial problem. In this paper we propose Information Maximising Autoencoder (InfoAE) where the encoder learns powerful disentangled representation through maximizing the mutual information between the representation and given information in an unsupervised fashion. We have evaluated our model on MNIST dataset and achieved 98.9 ($\pm$.1) % test accuracy while using complete unsupervised training.

## 1  INTRODUCTION

Learning disentangled representation from any unlabelled data is an active area of research (Goodfellow et al. (2016)). Self supervised learning (Gidaris et al. (2018); Zhang et al. (2016); Oord et al. (2018)) is a way to learn representation from the unlabelled data but the supervised signal is needed to be developed manually, which usually varies depending on the problem and the dataset. Generative Adversarial Neural Networks (GANs) (Goodfellow et al. (2014)) is a potential candidate for learning disentangled representation from unlabelled data (Radford et al. (2015); Karras et al. (2017); Donahue et al. (2016)). In particular, InfoGAN (Chen et al. (2016)), which is a slight modification of the GAN, can learn interpretable and disentangled representation in an unsupervised fashion. The classifier from this model can be reused for any intermediate task such as feature extraction but the representation learned by the classifier of the model is fully dependent on the generation of the model which is a major shortcoming. Because if the generator of the InfoGAN fails to generate any data manifold, the classifier is unable to perform well on any sample from that manifold. Tricks from Mutual Information Neural Estimation paper (Belghazi et al. (2018)) might help to capture the training data distribution, yet learning all the training data manifold using GAN is a challenge for the research community (Goodfellow et al. (2016)). Adversarial autoencoder (AAE) (Makhzani et al. (2016)) is another successful model for learning disentangled representation. The encoder of the AAE learns representation directly from the training data but it does not utilize the sample generation power of the decoder for learning the representations. In this paper, we aim to address this challenge. We aim to build a model that utilizes both training data and the generated samples and thereby learns more accurate disentangled representation maximizing the mutual information between the random condition/information and representation space.

## 2  INFORMATION MAXIMIZING AUTOENCODER

InfoAE consists of an encoder $E$, a decoder $D$ and a generator $G$. $G$ network produces latent variable space from a random latent distribution and a given condition/information. $D$ is used to generate samples from the latent variable space generated by the generator. It also maximizes the mutual information between the condition and the generated samples. $E$ is forced to learn the mapping of the train samples to the latent variable space generated by the generator. The model has three other networks for regulating the whole learning process: a classifier $C$, a discriminator $D_i$ and a self critic $S$. Figure 1 shows the architecture of the model.

## 2.1 ENCODER AND DECODER

The encoding network $E$, takes any sample $x \in p(x)$, where $p(x)$ is the data distribution. $E$ outputs latent variable $z_e = E(x)$, where $z_e \in q(z)$ and $q(z)$ can be any continuous distribution learned by $E$. This $z_e$ is feed to decoder network $D$ to get sample $\hat{x}_r$ so that $\hat{x}_r \in p(x)$ and $\hat{x}_r \approx x$.

## 2.2 GENERATOR AND DISCRIMINATOR

Generator network $G$ generates latent variable $z_g = G(z, c) \in p(z)$, from any sample $z$ and $c$ where $z \in u(z)$ and $c \in Cat(c)$. Here $p(z)$ can be any continuous distribution learned by $G$, $u(z)$ is random continuous distribution (e.g., continuous uniform distribution) and $Cat(c)$ is random categorical distribution. To validate $z_g$, decoder $D$ learns to generate sample $\hat{x}_g = D(G(z_g, c))$ so that $\hat{x}_g \in p(x)$. The discriminator network $D_i$ forces decoder to create sample from the data distribution.

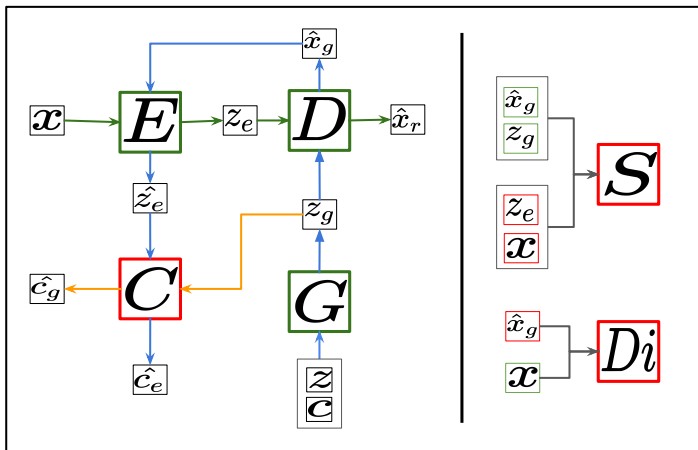

Figure 1: The architecture of the InfoAE, where $E$, $D$, $G$, $C$, $S$ and $D_i$ are encoder, decoder, generator, classifier, self critic and discriminator, respectively. The right section of the figure shows the discriminating networks $D_i$ and $S$ where the green and red boxes shows the true and false sample respectively.

## 2.3 CLASSIFIER AND SELF CRITIC

While generator generates $z_g$ from $z$, it can easily ignore the given condition $c$. To maximise the Mutual Information (MI) between $c$ and $z_g$, we use classifier network $C$ to classify $z_g$ into $\hat{c}_g = C(G(z_g, c))$ according to the given condition $c$. We also want encoder $E$ to learn encoding $\hat{z}_e = E(\hat{x}_g)$ so that $\hat{z}_e \in p(z)$ and $MI(\hat{z}_e, c)$ is maximised. To ensure $MI(\hat{z}_e, c)$ is maximised again the classifier network $C$ is utilised to classify $\hat{z}_e$ into $\hat{c}_e = C(\hat{z}_e)$ according to given condition $c$. To make sure $q(z) \approx p(z)$, we use a discriminator network $S$, which forces $E$ to encode $x$ into $p(z)$. $S$ learns through discriminating $(x, z_e)$ as fake and $(\hat{x}_g, z_g)$ as real sample. We named this discriminator as Self Critic as it criticises two generations from the sub networks of a single model where they are jointly trained.

## 2.4 TRAINING OBJECTIVES

The InfoAE is trained based on multiple losses. The losses are : Reconstruction loss, $R_l = \sqrt{\sum (\hat{x}_r - x)^2}$ for both Encoder and Decoder; Discriminator loss, $D_{il} = \log D_i(x) + \log(1 - D_i(D(z_g)))$ ; Decoder has loss $D_{lg}$, for the generated image $\hat{x}_g$ and loss $D_{le}$ for the reconstructed image $\hat{x}$, where $D_{lg} = \log(1 - D_i(D(z_g)))$ and $D_{le} = \log(1 - D_i(D(E(x))))$ ; Encoder loss $E_l = \log(1 - S(z_e, x))$ ; Self Critic loss $S_l = \log S(z_g, \hat{x}_g) + \log(1 - S(z_e, x))$ ; Two classification losses $C_{lg}$, $C_{le}$ respectively for Generator and Encoder where $C_{lg} = -\sum c \log(\hat{c}_g)$ and

$C_{le} = -\sum c \log(\hat{c}_e)$. We get our total loss, $T_l$ in equation 1 where $\alpha$, $\beta$ and $\gamma$ are hyper parameters.

$$T_l = \alpha * (C_{lg} + C_{le}) + \beta * (E_l + D_{le} + D_{lg}) + \gamma * R_l \tag{1}$$

All the networks are trained together and the weights of the $E$, $D$, $C$, and $G$ are updated to minimise the total loss, $T_l$ while the weights of the $S$ and $D_i$ are updated to maximise the loss $S_l$, $D_{il}$, respectively. So the training objective can be express by the equation 2

$$\min_{E,D,C,G} \max_D V(Di, S) = T_l + D_{il} + S_l \tag{2}$$

## 3 IMPLEMENTATION DETAILS

Our model has different components as shown in Figure 1. We used Convolutional Neural Network (CNN) for $E$, $D_i$ and $S$. Batch Normalization (Ioffe & Szegedy (2015)) is used except for the first and the last layer. We did not use any maxpool layer and the down sampling is done through increasing the stride. For classifier $C$ and generator $G$ we used simple two layers feedforward network with hidden layer. For Decoder $D$ we used Transpose CNN.

Our experiments show that the training of the whole model is highly sensitive to $\alpha$, $\beta$ and $\gamma$. After experimenting with different values of $\alpha$, $\beta$ and $\gamma$, we received best result for $\alpha = 1$, $\beta = 1$ and $\gamma = 0.4$. For $c$ variable we used random one hot encoding of size $10(c \sim \text{Cat}(K = 10, p = 0.1))$ and $z \in \mathbb{R}^{100}$, which is randomly sampled from a uniform distribution $U(-1, 1)$. The weights of all the networks are updated with Adam Optimizer (Kingma & Ba (2014)) and the learning rate of 0.0002 is used for all of them.

## 4 RESULTS AND DISCUSSION

We have evaluated the model on MNIST dataset and received outstanding results. InfoAE is trained on MNIST training data without any labels. After trainning, We encoded the test data with Encoder, $E$ and got classification label with the Classifier, $C$. Then we clustered the test data according to label and received classification accuracy of 98.9 ($\pm.05$), which is better than the popular methods as shown in Table 1.

Table 1: Comparison of Unsupervised Classification error rate of different models on MNIST test dataset.

| MODEL | ERROR RATE |
|---|---|
| InfoGAN (Chen et al. (2016)) | 5 |
| Adversarial Autoencoder (Makhzani et al. (2016)) | 4.10 ($\pm$ 1.12) |
| Convolutional CatGAN (Springenberg (2015)) | 4.27 |
| PixelGAN Autoencoders (Makhzani & Frey (2017)) | 5.27 ($\pm$ 1.81) |
| InfoAE | **1.1** ($\pm$ .1) |

The latent variable produced by the encoder on test data is visualized in figure 2(b). For visualization purpose we reduced the dimension of the latent vector with T-distributed Stochastic Neighbor Embedding or t-SNE (Van der Maaten & Hinton (2008)). In the visualization, we can observe that representation of similar digits are located nearby in the 2D space while different digits. This suggests that the encoder was able to disentangle the digits category in the representation space, which has eventually resulted in the superior performance. Also, the generator was able to generate latent space according to the condition and the decoder was able to generate samples from that latent variable space, disentangling the digit category as shown in Fig. 2(a).

Let us consider two latent variables $z_1 = E(x_1)$ and $z_2 = E(x_2)$ where $x_1$, $x_2$ are two sample images from the test data. Now let us do a linear interpolation between $z_1$ toward $z_2$ with $z_1 + (s/n) * (z_2 - z_1)$ where $s \in \{1, 2, ...., n\}$ and $n$ is the number of steps and feed the latent variables to the Decoder for generating sample. Figure 2(c) shows the interpolation between the samples from

different category. A smooth transition between different types of digits suggest the latent space is well connected.

Figure 2(d) show the interpolation between the same category of the samples and we can observe that the encoder was able to disentangle the styles of the digits in the latent space. This same category interpolation can be used as data augmentation.

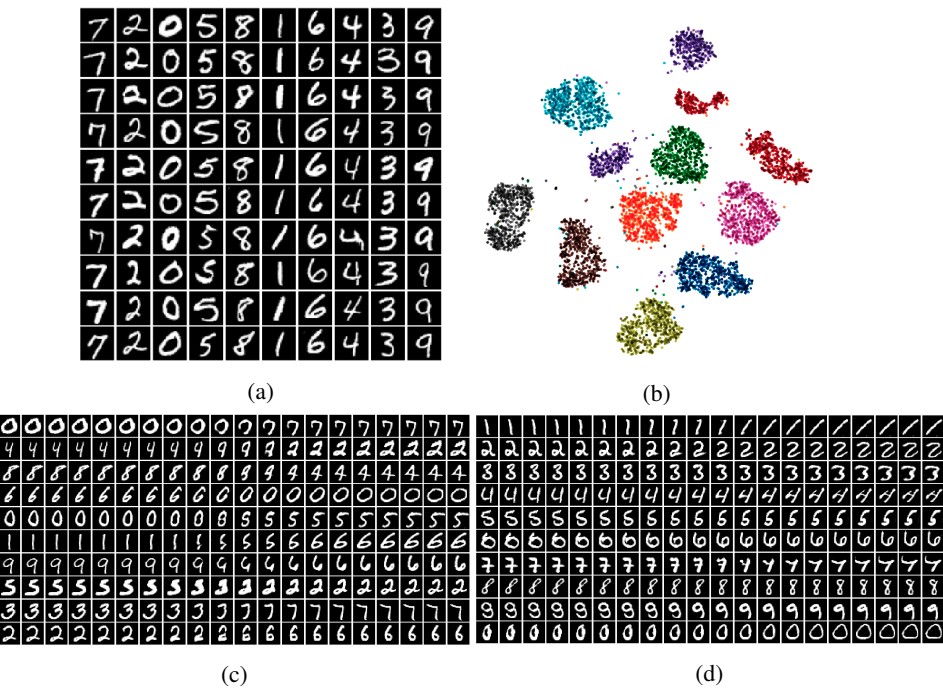

Figure 2: (a) Generated samples from the decoder. Rows show the samples $D(G(z, c))$ generated for different latent variables $z$ and columns show the samples generated for categorical variables $\{c_1, c_2, ......c_{10}\}$. (b) Representation of latent variables for the MNIST test data. Different colors indicate different category of the digits. (c) Visualization of linear interpolation between two reconstructed test samples from different categories (left to right). (d) Visualization of linear interpolation between two reconstructed test samples of the same category (left to right).

## 5    CONCLUSION AND FUTURE WORK

In this paper we present and validate InfoAE, which learns the disentangled representation in a completely unsupervised fashion while utilizing both training and generated samples. We tested InfoAE on MNIST dataset and achieved test accuracy of 98.9 ($\pm$.1), which is a very competitive performance compared to the best reported results including InfoGAN. We observe that the encoder is able to disentangle the digit category and styles in the representation space, which results in the superior performance. InfoAE can be used to learn representation from unlabelled dataset and the learning can be utilized in a related problem where limited labeled data is available. Moreover, its power of representation learning can be exploited for data augmentation. This research is currently in progress. We are currently attempting to mathematically explain the results. We are also aiming to analyze the performance of InfoAE on large scale audio and image datasets.

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
