# OpenReview forum: "Disentangled Representation Learning with Information Maximizing Autoencoder"
_ICLR.cc/2019/Workshop/LLD — Submitted to LLD 2019_

### Official Review · AnonReviewer1 · 2019-04-03
**Nice contribution, Some clarifications on the main claim and terminology are needed**

**Rating:** 4
**Confidence:** 3

**Review:**

The authors present a framework in which an auto encoder (E, D) is regularized such that its latent representation to share mutual information with a generated latent space representation. This generated latent representation is an output of a generator network (G) that produces its latent variables based on a given random noise variable (z) plus a categorical input (c). The mutual information is maximized using another network (C, classifier network) that tries to map generated latent representations  to the categorical input (c) of the generator network. Moreover two discriminator networks (S, D_i) are used to ensure that the generated images come from the same distribution as the data.

To the best of my understanding the presented scheme performs, in an unsupervised manner, a clustering to a pre-defined number of clusters, that are defined by the one hot encoding categorical (c) input of G, while at the same time G disentangles the inter-cluster variability to the random noise variable (z).

The language of the text is clear and the methodology is described in details. I have however a few comments concerning the presentation of the main claim.

1. According to the description of the proposed scheme, I find it difficult to understand how a disentangled representation is learned by the encoder (z_e). I understand that it is possible to generate a latent representation using the G network with c and z as inputs, yet this does not prove that the generated representation z_g (~z_e) are also disentangled.

2. Unfortunately the experimental setup does also not support the disentanglement claim in the auto-encoder representation space. An experiment on how the generated image looks when interpolating samples within the auto-encoder representation would be more insightful.

---

### Official Review · AnonReviewer2 · 2019-04-05
**Review: Disentangled Representation Learning with Information Maximizing Autoencoder**

**Rating:** 1
**Confidence:** 2

**Review:**

The introduction lacks some exposition/definition of "disentanglement" in the context of representations. Why are disentangled representations good?

Please remove the nested parentheses within the citations within parentheses (or change the citation style to one that doesn't have nested parentheses).

Figure 1 is a difficult to read. It would be easier if each component/input/output were labeled with the name of the component, rather than the variable. It also isn't clear from the figure which variables exist in the "latent variable space" (embedding?).

The notation p(x) almost always denotes a probability (i.e. a real value between 0 and 1) and so the notation x \in p(x) is somewhat confusing, since I assume x is meant to be an event in the probability space whose probability (function/density) is given by p(x).

Minor: In the variables with hats, the hats are off-center. To avoid this, use \hat{x}_r instead of \hat{x_r}.

It is also unclear what the relationship is between p(x), p(z), u(z), and q(z). Do p(x) and p(z) refer to the same distribution? This makes it very difficult for the reader to understand your approach.

Is the discriminator D_i indexed by i? Or is there only one discriminator? If there is only one discriminator, do not use subscript i as the variable name, since subcript i is almost always used as an index.

There is very little to motivate your architecture. Why have a self-critic?

What is V in equation 2? Same comment as above on the variable names in the loss expressions: it is easy to confuse l in S_l (among others) as an index.

How do you perform classification if you train on completely unlabeled data? The alternate methods you cite perform semi-supervised training.

Grammatical errors detract from the reader's ability to easily understand the content. For example, many nouns are missing determiners (e.g. should be "representations" or "a/the representation", "the G network", etc), and/or they should be plural (e.g. "adversarial autoencoders"). There are also some spelling errors (e.g. "training").

Minor: "Mutual information" need not be capitalized. Same with "convolutional neural network", "batch normalization", etc.

The bibliography items are inconsistently formatted.

---

### Decision · Program_Chairs · 2019-04-16
**Acceptance Decision**

**Decision:**

Reject

**Comment:**

The reviewers found a number of issues in clarity and were not fully convinced of the experiments